# Semantics-Aware Image Aesthetics Assessment using Tag Matching and Contrastive Ranking

## ABSTRACT

The perception of image aesthetics is built upon the understanding of semantic content. However, how to evaluate the aesthetic quality of images with diversified semantic backgrounds remains challenging in image aesthetics assessment (IAA). To address the dilemma, this paper presents a semantics-aware image aesthetics assessment approach, which first analyzes the semantic content of images and then models the aesthetic distinctions among images from two perspectives, i.e., aesthetic attribute and aesthetic level. Concretely, we propose two strategies, dubbed tag matching and contrastive ranking, to extract knowledge pertaining to image aesthetics. The tag matching identifies the semantic category and the dominant aesthetic attributes based on predefined tag libraries. The contrastive ranking is designed to uncover the comparative relationships among images with different aesthetic levels but similar semantic backgrounds. In the process of contrastive ranking, the impact of long-tailed distribution of aesthetic data is also considered by balanced sampling and traversal contrastive learning. Extensive experiments and comparisons on three benchmark IAA databases demonstrate the superior performance of the proposed model in terms of both prediction accuracy and alleviating long-tailed effect. The code of the proposed method will be public.

## CCS CONCEPTS

• **Computing methodologies → Image representations**.

## KEYWORDS

Image aesthetics assessment, Semantic and aesthetic perception, CLIP, Contrastive learning

## 1 INTRODUCTION

The rise of mobile Internet and the widespread popularity of social media platforms, such as Instagram and WeChat, have transformed the way we share and consume images. In this era, users have become increasingly focused on the aesthetic appeal of images. In view of this, image aesthetics assessment (IAA) has witnessed a notable surge in research interest in recent years, attracting a growing number of scholars dedicated to investigating its multifaceted dimensions. IAA approaches have extensive applications [5], such as image recommendation [39], image enhancement [8], and image retrieval [44], etc.

*ACM MM, 2024, Melbourne, Australia*

© 2024 Copyright held by the owner/author(s). Publication rights licensed to ACM.
ACM ISBN 978-x-xxxx-xxxx-x/YY/MM
https://doi.org/10.1145/nnnnnnn.nnnnnnn

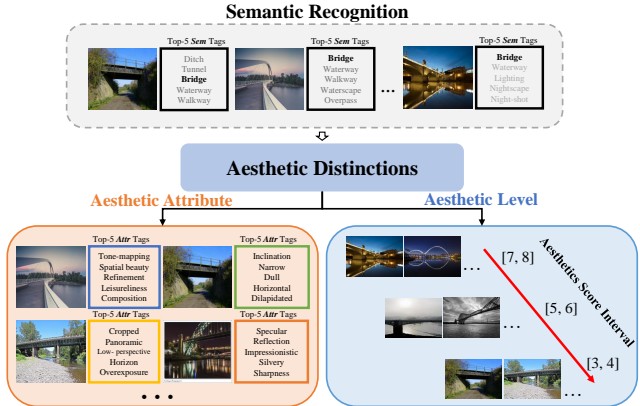

**Figure 1: Illustration of our method. Modeling aesthetic distinctions among images with similar semantic backgrounds from two aspects: aesthetic attribute and aesthetic level.**

In the past few years, a proliferation of novel IAA models have been reported in the literature. Early IAA models primarily used hand-crafted features and simple machine learning methods to classify images into high and low aesthetic categories, which were inspired by people's intuitive judgments of image aesthetics and widely accepted photography rules [4, 19]. With the boom of deep learning, the mainstream methods for aesthetics assessment started leveraging Convolutional Neural Networks (CNN) to unearth common aesthetic criteria embedded within large datasets. Accordingly, the IAA has been refined into three specific tasks: aesthetic binary classification [22, 41], aesthetic score regression [21, 24], and aesthetic distribution prediction [3, 40]. More recently, the emergence of large language models like GPT [37] has fueled the rapid development of the multimodal vision field. The IAA community has also made attempts on investigating the subsidiary effect of the language modality [7, 10, 18, 38].

The perception of image aesthetics is built upon the understanding of semantic content. Numerous existing IAA methods have proven the advantages of integrating semantic content into aesthetic perception modeling. There are two main types of integration: one involves incorporating semantic categories as additional labels for training in a multi-task manner [26, 41], while the other directly utilizes semantic features (e.g., features from pre-trained models) as the foundation for learning aesthetic features [6, 13]. These approaches are designed based on an assumption that aesthetic perception can be more easily achieved among images with similar semantic contents. However, there is no explicit comparison made among these images, so the model may not effectively discern and perceive aesthetic distinctions within the same semantic context. The aesthetic distinctions among images are typically determined

by two key factors. Firstly, the inherent aesthetic attributes of images, encompassing elements like composition (e.g., balanced or unfocused) and lighting (e.g., exposure or evenness), play a pivotal role in shaping its overall aesthetic quality. Secondly, the distinctions are directly reflected in the distribution of images with similar semantic backgrounds but different aesthetic levels. Analyzing the comparative relationships among these images helps to evaluate how well each image aligns with the aesthetic expectations and preferences associated with its semantic background.

Inspired by the above facts, this paper presents a semantics-aware image aesthetics assessment approach, which first analyzes the semantic content of images and then models aesthetic distinctions from two aspects: aesthetic attribute and aesthetic level, as illustrated in Figure 1. Concretely, we propose two strategies, dubbed tag matching and contrastive ranking, to extract knowledge pertaining to image aesthetics. In view of the differences in the lexical representation of semantic and aesthetic attribute in aesthetic description, two lexical tag libraries for semantic recognition and aesthetic attribute analysis are built. The tag matching then extracts similarity matching relationship between individual images and the predefined semantic and attribute tag libraries. The contrastive ranking is designed to uncover the comparative relationships between image sets with different aesthetic levels within specific semantic backgrounds.

The contributions of this work are threefold:

- We propose a semantics-aware image aesthetics assessment approach using tag matching and contrastive ranking (TMCR), which first analyzes the semantic content of images and then models aesthetic distinctions from two perspectives: aesthetic attributes and aesthetic levels.
- We design a novel contrastive ranking method to investigate the information about aesthetic levels from images under similar semantics, which extracts both intra-level aesthetic criteria and inter-level aesthetic ordering. Moreover, the problem of the long-tailed distribution is alleviated through balanced sampling and traversal contrastive learning.
- We conducted extensive experiments on three benchmark databases for image aesthetics assessment, including AVA, AADB, and PARA. The experimental results demonstrate that the proposed TMCR model outperforms the state-of-the-art methods and exhibits better long-tailed prediction capability.

## 2 RELATED WORK

### 2.1 Image Aesthetics Assessment

Early IAA models utilized hand-crafted features and simple machine learning to classify images into high and low aesthetic categories, based on photography rules. Datta *et al.* [4] extracted nine features from images, such as exposure of light and colorfulness, composition of regions and depth-of-field, based on which the Support Vector Machine (SVM) is utilized to build the aesthetic classification model. Ke *et al.* [19] employed perceptual features to classify high and low aesthetics images, including spatial distribution of edges, color distribution, and hue count, etc. Although these features have clear physical interpretation, it is challenging to comprehensively capture the aesthetic characteristics of images, due to the inherent

abstraction of image aesthetics and the limited understanding of the underlying mechanisms of aesthetics.

With the rise of deep learning, IAA methods began to adopt deep neural networks to uncover the common aesthetic criteria present in large datasets. Meantime, the IAA has been refined into three specific tasks: aesthetic binary classification, aesthetic score regression, and aesthetic distribution prediction. Murray *et al.* [33] first established a large-scale database for aesthetic visual analysis, called AVA. Recognizing the importance of both the global and local characteristics of images in aesthetic assessment, several approaches based on patch-level analysis have been developed [30, 31]. Lu *et al.* [29] proposed the RAPID algorithm, which leverages a double-column convolutional network to separately handle the global and local views of images for aesthetic classification. There are also some theme-oriented IAA networks, motivated by the fact that perception of image aesthetics is built upon the semantic understanding of content [21, 41]. He *et al.* [6] presented a network that combines aesthetic attributes and color space to jointly predict aesthetic distributions. Li *et al.* [26] proposed a theme-aware visual attribute reasoning framework, which utilizes a bilevel Graph Convolutional Network (GCN) to simulate the human aesthetic perception process. More recently, Jia *et al.* [13] designed an effective full-resolution technique to perform the IAA task based on theme information.

Currently, with the rise of multimodal learning, aesthetic-related information from the language modality has been introduced into IAA. Large pre-trained vision-language models, like CLIP [36], have been proved to be more suitable for image aesthetics modeling, because the language modality contains rich semantics and aesthetics clues [7]. Zhou *et al.* [52] constructed the AVA-Comments database, the first multimodal database for IAA that combines visual and textual information. They also proposed a multimodal IAA model based on Deep Boltzmann Machine (DBM) for binary aesthetic classification. Ke *et al.* [18] pretrained an image-text encoder-decoder model using contrastive and generative objectives to learn multimodal aesthetic representations from user comments. Sheng *et al.* [38] proposed an attribute-aware contrastive learning strategy to mitigate the domain shift from the general visual domain to the aesthetics domain, improving CLIP's performance on the IAA task. These studies have provided compelling evidence that the integration of textual information can greatly enhance the representation capability of IAA models.

### 2.2 Contrastive Learning

Contrastive learning [23] is a representation learning technique that compares similar and dissimilar data pairs to learn meaningful patterns and discriminative information. It encourages the model to group similar instances together and separate dissimilar ones, resulting in effective feature representations. Wu *et al.* [43] proposed 'Instance Discrimination', which treats two augmented views of an image as a positive pair and other images as negative pairs, aiming to train the model to learn instance-specific features invariant to different image enhancements. Li *et al.* [25] introduced a method that combines prototypical networks with contrastive learning for unsupervised representation learning. By integrating prototypical

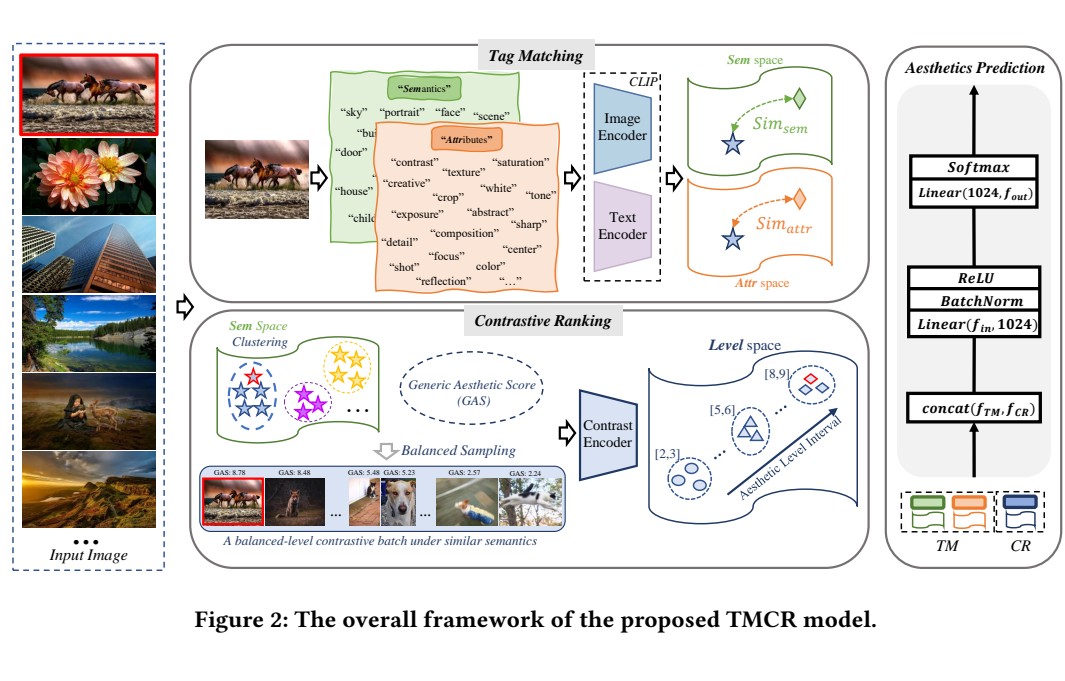

**Figure 2: The overall framework of the proposed TMCR model.**

information, the model learns more discriminative and semantically meaningful representations, leading to improved performance in downstream tasks. Contrastive learning has been successfully applied in various domains, allowing models to capture and utilize valuable insights from the data [47, 49].

In the area of quality perception, Chen *et al.* [2] applied contrastive learning to video quality assessment, using distortion types as positive and negative selection criteria to learn effective representations for video quality score regression. Zhao *et al.* [50] developed diverse degradation strategies to generate distorted images and utilized content-aware and quality-aware comparisons to derive representations related to image quality. Contrastive learning also has great potential for addressing long-tailed problems. Kang *et al.* [16] discovered that the self-supervised contrastive learning methods perform stably well even when the datasets are heavily imbalanced. They further introduced a K-positive contrastive learning that effectively combines the advantages of supervised learning and contrastive learning to achieve balanced and discriminative representations. Zhu *et al.* [53] introduced the Balanced Contrastive Learning (BCL) method, which utilizes class-averaging and class-complement strategies to address the optimization bias in Supervised Contrastive Learning (SCL) [20] and enhance the performance of long-tailed visual recognition tasks. These studies provide insights into the investigation of contrastive learning to explore the aesthetic comparative relationship and mitigate the impacts of the long-tailed distribution of aesthetic data.

## 3 PROPOSED METHOD

### 3.1 Problem Formulation

Image aesthetics assessment (IAA) aims at mining the general public's criteria of aesthetics from large-scale image aesthetics data, and then generating aesthetic evaluation $\hat{y}$ in line with the ground-truth aesthetic quality of a given image $I$, which can be formulated

as:

$$\hat{y} = \mathcal{M}\left(I | \boldsymbol{Aes}_G \longleftarrow \theta\right), \qquad (1)$$

where $\mathcal{M}(\cdot)$ represents the established IAA model with $\theta$ as the parameter, $\boldsymbol{Aes}_G$ refers to the generic aesthetics consensus learned from large-scale aesthetic annotation data.

The perception of aesthetics is grounded in semantic analysis, and previous studies have shown the benefits of utilizing semantic analysis for modeling aesthetic perception [6, 11, 13, 21, 26, 41]. In this study, we specifically examine the connection and integration of semantic analysis to aesthetic attributes and levels. This is accomplished by the proposed tag matching (TM) and contrastive ranking (CR) strategies. By incorporating these two relationships, we train the IAA model to generate aesthetic predictions. The whole process can be expressed as:

$$\hat{y} = \mathcal{M}\left(I | \boldsymbol{Aes}_{sem}^{attr} \longleftarrow \theta_{TM}, \boldsymbol{Aes}_{sem}^{level} \longleftarrow \theta_{CR}\right), \qquad (2)$$

where, $\boldsymbol{Aes}_{sem}^{attr}$ and $\boldsymbol{Aes}_{sem}^{level}$ represent the prior knowledge related to aesthetic perception modeling, obtained from parameters $\theta_{TM}$ and $\theta_{CR}$, respectively. Figure 2 shows the overall framework of the proposed TMCR model.

### 3.2 Tag Matching

When describing an image in aesthetic perception, aesthetic description and semantic description are often intertwined, and aesthetic description cannot exist independently from semantics. Additionally, words describing semantics are often associated with objects or themes, while words describing aesthetics are often related to attributes such as clarity, composition, and feeling. Recently, Zhong *et al.* [51] conducted manual selection of object-related words and aesthetics-related words to calculate the Aesthetic Relevance Score (ARS) of a sentence, which has contributed to the development of aesthetics assessment and aesthetics captioning. Building upon the observations and previous studies, we construct two lexical tag libraries, $Tag_{sem}^M$ and $Tag_{attr}^N$, for semantic recognition and

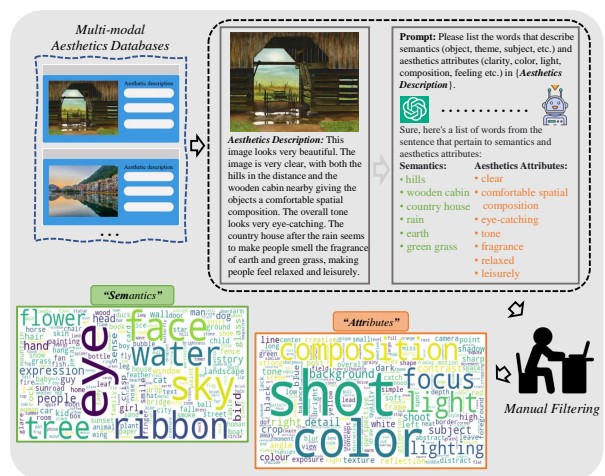

**Figure 3: The process of constructing semantic and attribute tag libraries.**

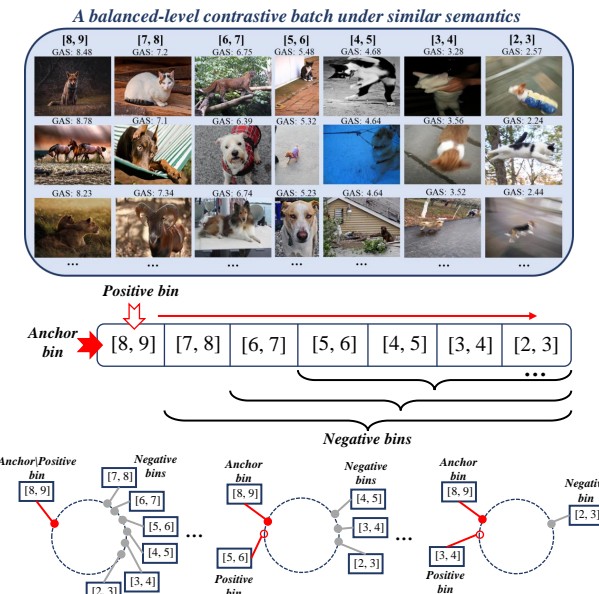

**Figure 4: Illustration of the contrastive ranking. A balanced-level batch is first sampled according to semantic contents and GAS. Then, pull-pushing force of contrastive learning is employed to explore intra-level aesthetic criteria and inter-level aesthetic ordering (Taking '[8, 9]' as an example).**

aesthetic attribute analysis. These libraries are built using existing multi-modal aesthetics databases, including DPC2022 [51], which is currently one of the largest datasets of aesthetic comments, and EPAD [12], which contains high-quality aesthetic comments from experts. Specifically, we utilize GPT4 and manual filtering to create these tag libraries, and the process is shown in Figure 3. Finally, we obtain 2145 semantic recognition tags and 1659 attribute analysis tags.

Next, we employ the multimodal pretrained model CLIP to identify the semantic content and aesthetic attributes of an image. Initially, the textual tags are converted into deep feature representations using the text encoder $E_T$ of CLIP, which is expressed as follows:

$$\mathcal{F}_{sem}^{m} = E_T\left(T_{sem}^{m}\right), T_{sem}^{m} \in Tag_{sem}^{M}, \tag{3}$$

$$\mathcal{F}_{attr}^{n} = E_T\left(T_{attr}^{n}\right), T_{attr}^{n} \in Tag_{attr}^{N}, \tag{4}$$

where $\mathcal{F}_{sem}^{m}$ and $\mathcal{F}_{attr}^{n}$ represent the transformed semantic tag feature and attribute tag feature, respectively. Given an image $I$, cosine similarity is utilized to establish the correlation between the image and the two tag libraries:

$$Sim_{sem}^{M} = \cos_{m \in M}\left(E_I\left(I\right), \mathcal{F}_{sem}^{m}\right), \tag{5}$$

$$Sim_{attr}^{N} = \cos_{n \in N}\left(E_I\left(I\right), \mathcal{F}_{attr}^{n}\right), \tag{6}$$

where $Sim_{sem}^{M}$ and $Sim_{attr}^{N}$ respectively represent the similarity features with two tag libraries. $E_I$ is the image encoder of CLIP. Finally, the direct concatenation of the two similarity features is utilized to represent the interconnection between semantic content and aesthetic attributes in an image:

$$\mathcal{F}_{sem}^{attr} = concat\left(Sim_{sem}^{M}, Sim_{attr}^{N}\right), \tag{7}$$

where, $\mathcal{F}_{sem}^{attr}$ represents the fused feature, and $concat\left(\cdot\right)$ denotes the feature concatenation.

## 3.3 Contrastive Ranking

The goal of contrastive ranking is to uncover the comparative relationships between image sets with different aesthetic levels but similar semantic backgrounds. The comparative relationships are reflected in the distribution of images within and across aesthetic levels, representing aesthetic criteria and aesthetic ordering, respectively. To this end, the images within an aesthetics database are first semantically clustered, and then divided into subsets according to their semantic categories and generic aesthetic scores (GAS). Furthermore, considering the long-tailed distribution problem of existing aesthetic data, the majority of images in the training set are primarily distributed in the middle score segment, and there is a relatively small number of images in the low and high score segments, resulting in poor prediction performance for high-quality and low-quality images. To address this issue, we employ a balancing process for the divided sets, generating relatively balanced batches for each aesthetic level across different semantic backgrounds. Then, the traversal contrastive learning is designed to explore intra-level aesthetic criteria and inter-level aesthetic ordering. The process of contrastive ranking is illustrated in Figure 4.

*Semantic Clustering.* Given an image aesthetics database $\mathcal{D}_{aes} = \left\{\left(I^i, y^i\right)\right\}_{i=1}^{N_{aes}}$, where $I^i$ denotes the i-th image, $y^i$ denotes the aesthetic distribution (which can be transformed into GAS) and $N_{aes}$ represents the number of images. Firstly, Eq. 5 is utilized to identify the semantic content of images based on their similarity to the predefined semantic tag library. Subsequently, the K-means algorithm is employed to cluster the images with similar semantic backgrounds. Within each cluster, the images are reorganized based

on different aesthetic levels. Currently, most databases use integer score ranges from 1 to 10 to collect aesthetic ratings. Therefore, at this stage, we also partition the data based on this rating level setting to obtain a new database $\mathcal{D}_c$ for subsequent training:

$$\mathcal{D}_c = \left\{ ... \left[ ..., (I_1, I_2, ...)^l, ... \right]^k, ... \right\}, \tag{8}$$

where, $[\cdot]^k$ denotes k-th semantic cluster, and $(\cdot)^l$ in the cluster represents the image set at l-th aesthetic level.

*Balanced Sampling.* Considering the uneven distribution of images across different aesthetic levels, it is crucial to mitigate the impact of this data distribution on modeling the contrastive relationship. Therefore, a balanced sampling process is conducted on the database $\mathcal{D}_c$ obtained from the previous step. Specifically, for a particular semantic cluster $[\cdot]^k$, a balanced quantity value $N_s$ is generated based on the distribution of images across different aesthetic levels within the cluster. Then, a random sampling operation is performed from the image sets for each aesthetic level, forming a balanced-level batch. Through repeated sampling, multiple balanced batches are obtained with different semantic backgrounds.

*Traversal Contrastive Learning.* For a balanced-level batch, the traversal contrastive learning is designed to explore the aesthetic comparative relationship among images distributed at different aesthetic levels. Concretely, for samples within the aesthetic level interval '[8, 9]', which is regarded as an 'Anchor bin', the traversal contrastive process can be described as follows. In the first-round comparison, the samples within the interval are considered as positive samples for each other, while the samples in other intervals in the batch are considered as negative samples. In the second-round comparison, the 'Positive bin' is adjusted to '[7, 8]', and all level intervals less than 7 are considered as 'Negative bins'. This comparison process repeats until all level intervals in the batch are traversed, which represents a traversal comparison of the anchor bin '[8, 9]'. The first-round comparison aims to learn intra-level aesthetic criteria and the subsequent comparisons aim to learn inter-level aesthetic ordering.

The contrastive encoder $E_{CR}$ is introduced to convert images into deep features, which is represented as: $\mathcal{F}_{sem}^{level} = E_{CR}(I_i)$, where, $\mathcal{F}_{sem}^{level}$ denotes the transformed deep feature. The contrastive loss based on the aesthetic level relationship of this batch is defined as:

$$\mathcal{L}_{CR} = \sum_{anchor} \sum_{pos} \Psi_c \left( anchor_{bin}, pos_{bin}, neg_{bin} \right), \tag{9}$$

$$\Psi_c = -\log \frac{\exp \left( \mathcal{F}^a * \mathcal{F}^p / \tau \right)}{\exp \left( \mathcal{F}^a * \mathcal{F}^{p,n} / \tau \right)}, \tag{10}$$

where, $\mathcal{L}_{CR}$ is the contrastive ranking loss, $anchor_{bin}, pos_{bin}, neg_{bin}$ represent the anchor sample interval and its corresponding positive and negative sample interval in the process of traversal contrastive learning, respectively. $\Psi_c$ is the contrastive loss, $a$, $p$, $n$ are samples from $anchor_{bin}, pos_{bin}, neg_{bin}$, $*$ represents the inner product operation, and $\tau$ is the temperature coefficient.

By updating the parameters through balanced-level batches sampled from different semantic clusters, the contrastive encoder $E_{CR}$ extracts knowledge related to aesthetic levels in image sets across various semantic contexts:

$$\theta_{CR} \longleftarrow \theta_{CR} - \alpha \cdot \frac{1}{N_b} \sum_{batch} \frac{\partial \mathcal{L}_{CR}}{\partial \theta_{CR}}, \tag{11}$$

where $\theta_{CR}$ is the parameter of the contrastive encoder, $\alpha$ is the learning rate, and $N_b$ is the number of balanced batches.

## 3.4 Aesthetics Prediction

Based on the modeling of aesthetic attribute and aesthetic level, aesthetics prediction is performed in this stage. Specifically, we perform dimension transformation on the $\mathcal{F}_{sem}^{attr}, \mathcal{F}_{sem}^{level}$, and then directly concatenate them:

$$\mathcal{F}_{aes} = concat \left( \mathcal{F}_{sem}^{attr}, \mathcal{F}_{sem}^{level} \right), \tag{12}$$

where, $\mathcal{F}_{aes}$ represents the fused feature. A score $MLP$ is used to generate aesthetic distribution $\hat{y}$ based on the fused feature:

$$\hat{y} = MLP \left( \mathcal{F}_{aes} \right). \tag{13}$$

The Earth Mover's Distance (EMD) loss is applied to optimize the model parameters:

$$\mathcal{L}_{aes} = \left( \frac{1}{D} \sum_i^D \left| CDF_y(d) - CDF_{\hat{y}}(d) \right|^r \right)^{\frac{1}{r}}, \tag{14}$$

$$\theta \left( E_{CR}, MLP \right) \longleftarrow \theta - \beta \cdot \frac{1}{N_{aes}} \sum_i \frac{\partial \mathcal{L}_{aes}}{\partial \theta}, \tag{15}$$

where, $\mathcal{L}_{aes}$ is the loss function, $CDF(\cdot)$ is the cumulative distribution function, $d$ represents the dimensions of the aesthetic distribution, $r$ is set to 2 to penalize the Euclidean distance. $\theta(E_{CR}, MLP)$ denotes the parameter to be updated, $\beta$ is the learning rate.

# 4 EXPERIMENTS

## 4.1 IAA Databases

To gauge the efficacy of the proposed IAA model, we conduct experiments on three benchmark IAA databases, including AVA [33], AADB [21], and PARA [45].

**AVA** (Aesthetic Visual Analysis) [33] is currently the largest and most widely used database in the field of IAA. The database contains a total of 255,530 images. Each image receives annotations from an average of 210 annotators, with scores ranging from 1 to 10. Researchers commonly employ a standardized data partitioning approach, using approximately 230,000 images for training and the remaining 20,000 images for testing.

**AADB** (Aesthetics and Attributes DataBase) [21] is a database specifically designed for analyzing aesthetic attributes of images. The database comprises 10,000 images, each annotated by at least five users. The annotations include 11 aesthetic attributes and an overall aesthetic rating for each image. The attribute scores range from -1 to 1, while the overall aesthetic rating ranges from 1 to 5.

**PAPA** (Personalized image Aesthetics database with Rich Attributes) [45] consists of 31,220 images annotated by 438 participants. It includes image-centric annotations such as overall aesthetic ratings (1-5), distortion quality ratings (1-5), scene categories (10 types), and aesthetic attributes (5 types). Desensitized subjective information is also recorded, including gender, age, education

Table 1: Comparison of the proposed TMCR model with the state-of-the-art methods for three IAA tasks: aesthetic binary classification, aesthetic score regression, and aesthetic distribution prediction, on the AVA database.

| Method | Backbone | Input Size | Classification ACC↑ | Score Regression | | | Distribution | |
|--------|----------|------------|------|------|------|------|------|------|
| | | | | PLCC↑ | SRCC↑ | MSE↓ | EMD1↓ | EMD2↓ |
| DMA-Net [30] | AlexNet | 227×227 | 75.4 | - | - | - | - | - |
| Kong et al. [21] | AlexNet | 227×227 | 77.3 | - | 0.558 | - | - | - |
| NIMA [40] | VGG16 | 224×224 | 78.2 | 0.647 | 0.633 | 0.330 | 0.049 | 0.071 |
| APM [32] | ResNet-101 | Resize (500) | 80.3 | - | 0.709 | 0.279 | - | 0.061 |
| A-Lamp [31] | VGG16 | 224×224 | 82.5 | - | - | - | - | - |
| Zeng et al. [48] | ResNet-101 | 384×384 | 80.8 | 0.720 | 0.719 | 0.275 | - | 0.065 |
| Hosu et al. [9] | InceptionResnet | Full resolution | 81.7 | 0.757 | 0.756 | - | - | - |
| MUSIQ [17] | ViT | Full resolution | 81.5 | 0.738 | 0.726 | 0.242 | - | - |
| Niu et al. [34] | ResNet-50 | 224×224 | 81.9 | 0.740 | 0.734 | 0.242 | - | - |
| TANet [6] | MobileNet-v2, ResNet18 | 224×224 | 80.6 | 0.765 | 0.758 | - | 0.047 | - |
| AesCLIP [38] | ViT | 224×224 | **83.1** | 0.779 | 0.771 | 0.218 | 0.041 | **0.058** |
| VILA [18] | ViT | 224×224 | - | 0.774 | 0.774 | - | - | - |
| Jia et al. [13] | Inception-V3 | Full resolution | 82.4 | 0.775 | 0.774 | 0.231 | **0.039** | - |
| TMCR | ResNet-50 | 224×224 | 81.7 | 0.760 | 0.753 | 0.231 | 0.047 | 0.068 |
| TMCR | ViT | 224×224 | 82.1 | 0.778 | 0.771 | 0.219 | 0.045 | 0.065 |
| TMCR | Swin-T(b) | 224×224 | 82.8 | **0.790** | **0.782** | **0.210** | 0.043 | 0.061 |

background, art and photography experience, personality traits, emotional responses, etc.

## 4.2 Experimental Settings

**Training Settings.** We implement the model using PyTorch [35] and utilize the AdamW [28] optimizer for parameter updating. In the tag matching stage, we employ the ViT-L/14 version of CLIP to extract similarity features. Notably, to preserve its matching ability, both the image and text encoders remain frozen throughout the process. In the contrastive ranking stage, IAA databases are reorganized based on semantic clustering and aesthetic levels. For aesthetics prediction, a score MLP consisting of two linear layers followed by Softmax is used. Regarding hyper-parameters, the learning rate $\alpha$ for contrastive ranking is set to 1e-5 with weight decay of 5e-2. The temperature coefficient $\tau$ is set to 0.07. The batch size depends on the distribution of images across different aesthetic levels. The training process lasts for 50 epochs. For aesthetics prediction, the learning rate $\beta$ is set to 1e-4 with a batch size of 64, decaying by 0.9 per epoch until convergence.

**Evaluation Metrics.** For aesthetic binary classification, Accuracy (ACC) is used to measure performance. For aesthetic score regression, Spearman Rank order Correlation Coefficient (SRCC), Pearson Linear Correlation Coefficient (PLCC) and Mean Squared Error (MSE) are used to quantify the consistency between the predicted scores and the ground-truth GAS. For aesthetic distribution prediction, EMD with $r = 1$ and $r = 2$ are calculated, which are represented as EMD1 and EMD2, respectively.

## 4.3 Performance Evaluation

**Performance on AVA.** We first conducted experiments to evaluate the performance of our model on the widely used AVA database. We compared our model with 13 IAA models, including both classic and the latest approaches, and the results are summarized in Table 1. It is observed from the table that the proposed TMCR shows excellent performance, particularly in aesthetic score regression, surpassing

Table 2: Comparison on AADB and PARA databases.

| Database | Method | SRCC↑ | PLCC↑ |
|----------|--------|-------|-------|
| AADB | NIMA [40] | 0.708 | 0.711 |
| | Hosu et al. [9] | 0.725 | 0.726 |
| | PA_IAA [27] | 0.720 | 0.728 |
| | TANet [6] | 0.738 | 0.737 |
| | MUSIQ [17] | 0.706 | 0.712 |
| | Celona et al. [1] | 0.757 | 0.762 |
| | TAVAR [26] | 0.761 | 0.763 |
| | TMCR | **0.775** | **0.773** |
| PARA | NIMA [40] | 0.886 | 0.923 |
| | Hosu et al. [9] | 0.842 | 0.892 |
| | PA_IAA [27] | 0.877 | 0.919 |
| | TANet [6] | 0.883 | 0.917 |
| | MUSIQ [17] | 0.882 | 0.918 |
| | Yang et al. [45] | 0.902 | 0.936 |
| | TAVAR [26] | 0.911 | 0.940 |
| | TMCR | **0.915** | **0.945** |

all previous algorithms. Furthermore, compared with the methods with full-resolution input, our approach utilizes cropped patches as input and still achieves notably superior performance. Moreover, with an relatively lightweight backbone of ResNet-50, the proposed method still delivers very competitive performance.

**Performance on AADB and PARA.** We further evaluated the performance of the proposed model on the AADB and PARA databases. These two databases have a relatively smaller number of raters per image compared to AVA. Consequently, the existing approaches primarily reported SRCC and PLCC metrics. Table 2 summarizes the performance of our model compared to the state-of-the-art IAA models on the AADB and PARA databases. The results demonstrate that our model achieves the best performance. Compared to methods trained with attribute labels contained in databases, such as TAVAR [26], the proposed model achieved even better performance by solely utilizing aesthetic score annotations.

**Table 3: Comparison of long-tailed prediction performance between the proposed model and state-of-the-art IAA methods on the balanced test set of AVA database.**

| Metrics | MSE↓ | | | | SRCC↑ | PLCC↑ |
|---|---|---|---|---|---|---|
| Shot | ALL | Many | Med. | Few | | |
| NIMA [40] | 0.432 | 0.144 | 0.347 | 0.978 | 0.859 | 0.851 |
| TANet [6] | 0.718 | **0.089** | 0.537 | 1.902 | 0.860 | 0.855 |
| AesCLIP [38] | 0.383 | 0.149 | 0.289 | 0.863 | 0.878 | 0.873 |
| VILA [18] | 0.685 | 0.572 | 0.672 | 0.870 | 0.879 | 0.870 |
| TMCR | **0.361** | 0.142 | **0.269** | **0.818** | **0.885** | **0.875** |

## 4.4 Long-Tailed Performance

Imbalanced data distributions, where certain target values have significantly fewer observations, are commonly encountered in real-world datasets. Existing techniques for handling imbalanced data primarily concentrate on categorical indices, typically seen in classification problems. Yang *et al.* [46] first introduced the concept of Deep Imbalanced Regression (DIR). In the IAA field, there is very limited research addressing the aesthetic deep imbalanced regression problem [14, 15]. However, aesthetic data naturally exhibits a long-tailed distribution, as show in Figure 5(a), and accurately discerning the tail-end data holds significant practical value.

In the proposed method, we designed two strategies, namely balanced sampling and traversal contrastive learning, to alleviate the issue of long-tailed distribution when extracting knowledge related to aesthetic levels. For long-tailed testing, we divide score ranges into three regions ('Many', 'Med', and 'Few') based on the distribution of image quantities in the AVA training set. We sample an equal number of samples from each corresponding region in the original test set, creating a balanced test set (457 images for each region) for long-tailed evaluation.

We compare the performance of the proposed TMCR model with NIMA [40] (considered as the baseline), TANet [6], AesCLIP [38], and VILA [18] (recent models with top performances) on the balanced test set. We reported the comparative results of relevant metrics in Table 3, where MSE is employed to quantify the prediction performance across different regions, SRCC and PLCC are used to measure the correlations across the whole score range. From Table 3, we have the following observations. First, all models experienced varying degrees of long-tailed impacts, manifested by a decrease in performance in the corresponding region of testing as the training data decreased. Secondly, the proposed TMCR achieves the best results in terms of overall SRCC and PLCC metrics, as well as MSE metrics across 'ALL', 'Med.' and 'Few' regions, which demonstrates the advantage of TMCR in long-tailed testing. Furthermore, in Figure 5(b), we visualize the actual prediction performance of these models. It can be observed that 'TANet' predominantly generates predicted scores within the middle range and 'VILA' exhibits the widest range of score prediction intervals. The higher MSE of TANet and VILA compared to NIMA suggests that TANet's predictions for the 'Few' region significantly affect the overall performance, while VILA exhibits severe prediction errors despite its ability to handle a wide range of scores. These findings highlight the importance of investigating the long-tailed problem in IAA.

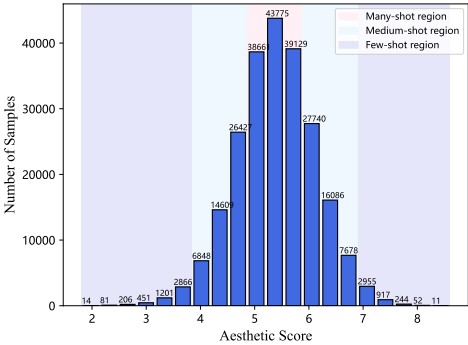

(a) Data distribution

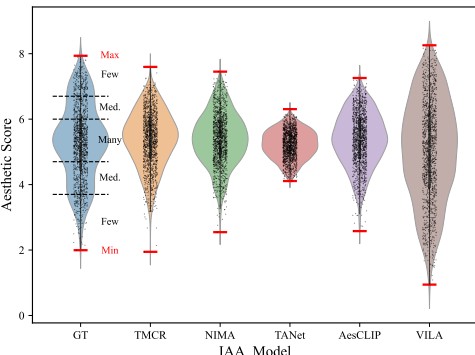

(b) Prediction performance

**Figure 5: Data distribution in AVA training set and prediction performance of different IAA models on AVA balanced testing set.**

**Table 4: Ablation study on the impact of TM and CR.**

| Method | Classification | Score Regression | | | Distribution | |
|---|---|---|---|---|---|---|
| | ACC↑ | PLCC↑ | SRCC↑ | MSE↓ | EMD1↓ | EMD2↓ |
| TM-Net | 80.3 | 0.732 | 0.721 | 0.251 | 0.047 | 0.068 |
| CR-Net | 81.2 | 0.745 | 0.733 | 0.251 | 0.051 | 0.073 |
| TM+CR_LP | 81.9 | 0.771 | 0.762 | 0.224 | 0.045 | 0.064 |
| TM+CR_FT | 82.8 | 0.790 | 0.782 | 0.210 | 0.043 | 0.061 |

## 4.5 Ablation Study

**Impact of TM and CR.** Table 4 summarizes the results obtained from the ablation study on the impact of TM and CR. Specifically, 'TM-Net' and 'CR-Net' represent aesthetics prediction based solely on tag matching and contrastive ranking, respectively. 'LP' and 'FT' denote the contrastive encoder under two commonly used evaluation protocols, namely linear probe and fine-tuning. The results demonstrate the validity of the two branching features for modeling image aesthetics.

**Impact of Semantics and Attribute Tags.** We also conduct a comparison between using semantic tags alone and using both semantics and aesthetic attribute tags during the tag matching stage, and the results are listed in Table 5. The results demonstrate that the

**Table 5: Ablation study on the impact of Semantics and Attribute tags.**

| Tags | Classification | Score Regression | | | Distribution | |
|------|------|------|------|------|------|------|
| | ACC↑ | PLCC↑ | SRCC↑ | MSE↓ | EMD1↓ | EMD2↓ |
| Sem-only | 82.1 | 0.780 | 0.770 | 0.223 | 0.046 | 0.065 |
| Sem+Attr | 82.8 | 0.790 | 0.782 | 0.210 | 0.043 | 0.061 |

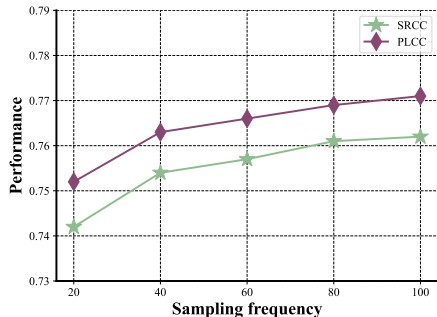

**Figure 6: Performances with different sampling frequency.**

use of semantic tags alone, delivers a very competitive performance. Incorporating attribute tags leads to further improvements to the model performance.

**Impact of Sampling Frequency.** We further conduct experiments to analyze the impact of sampling frequency on the learned representations during contrastive ranking. The contrastive ranking is performed on balanced-level batches sampled from different semantic clusters. Figure 6 shows the model performance in the linear probe protocol with varying sampling frequencies. The results clearly demonstarte that increasing the sampling frequency results in enhanced feature representations.

## 4.6 Visual Analysis

The proposed IAA model aims to capture the aesthetic distinctions among images with similar semantic backgrounds from two aspects: aesthetic attribute and aesthetic level. To visually demonstrate the effectiveness of the model in representation difference across the Semantic space, Attribute space and Level space, we conduct a visual analysis. Specifically, we utilize the T-SNE [42] to visualize the feature embeddings $\mathcal{F}_{sem}^{m}$ of individual images in relation to the semantic tag library. Subsequently, we visualize the attribute feature embeddings $\mathcal{F}_{attr}^{n}$ and the level feature embeddings $\mathcal{F}_{sem}^{level}$ of images from a balanced-level batch, which is sampled from a specific semantic cluster. In addition, we visualize three instances, their top-5 semantic and aesthetic attribute tags, as well as their mappings in three spaces. The visualizations of *Sem* space, *Attr* space and *Level* space are shown in Figure 7. It can be observed that, features displaying clustering patterns in the semantic space show distinct separability in both the attribute space and the level space. Specifically, in the level space, features belonging to different aesthetic levels are clearly separated and arranged according to their hierarchical relationship. These findings provide evidence for the effectiveness of the tag matching and contrastive ranking

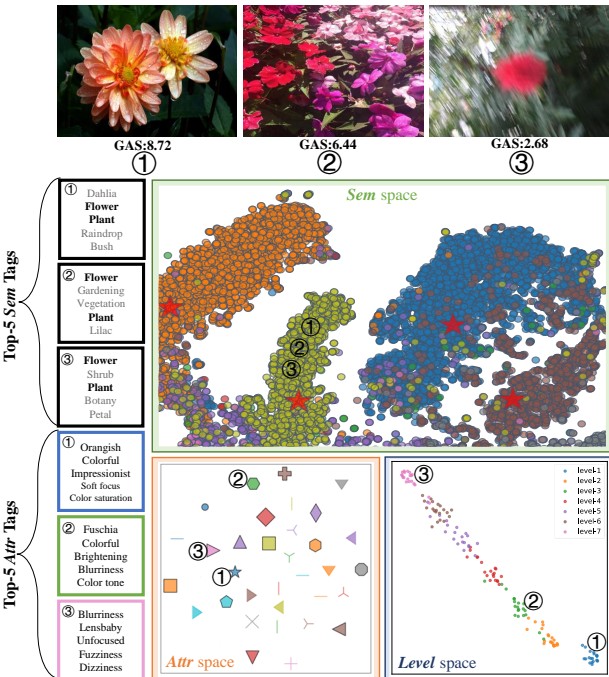

**Figure 7: Visualizations of feature embeddings of images in *Sem* space, *Attr* space and *Level* space, and three images from a specific cluster (ID: ①②③) with their top-5 semantic and aesthetic attribute tags, as well as their mappings in three spaces.**

strategies, thus demonstrating the efficacy of the proposed TMCR in modeling aesthetic perception.

## 5 CONCLUSION

In this paper, we have presented a semantics-aware image aesthetics assessment approach using tag matching and contrastive ranking, dubbed TMCR. We learn effective representations in the aesthetic attribute space and aesthetic level space to model the aesthetic distinctions among images under similar semantic backgrounds. In view of the differences in the expression of semantics and aesthetic attribute in aesthetic description, we propose leveraging CLIP's text-image matching ability to model the relationship between individual images and two predefined tag libraries. Moreover, we proposed a novel contrastive ranking strategy to model aesthetic level knowledge. During the contrastive ranking process, the impact of the long-tailed distribution of aesthetic annotation data is alleviated through balanced sampling and traversal contrastive learning. Extensive experimental results demonstrate that our proposed model outperforms state-of-the-art methods. While very encouraging performances have been achieved in this work, further investigations are needed to explore the extraction and modeling of aesthetic-related information in the language modality, as well as addressing the challenge of aesthetic deep imbalanced regression problem.

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
