# OpenReview forum: "Semantics-Aware Image Aesthetics Assessment using Tag Matching and Contrastive Ranking"
_acmmm.org/ACMMM/2024/Conference — MM2024 Poster_

### Official Review · Reviewer_wQLV · 2024-05-19

**Rating:** 5
**Confidence:** 4

**Summary:**

Semantic Content Analysis: The approach begins by analyzing the semantic content of images to establish an understanding of their meaning and context.
Modeling Aesthetic Distinctions: It models the aesthetic differences among images from two perspectives:
Aesthetic Attribute: Identifying the dominant aesthetic attributes based on predefined tag libraries through a strategy called "tag matching."
Aesthetic Level: Uncovering comparative relationships among images with similar semantic backgrounds but different aesthetic levels using "contrastive ranking."
Consideration of Data Imbalance: During contrastive ranking, the impact of long-tailed distribution in aesthetic data is addressed using balanced sampling and traversal contrastive learning to ensure accurate assessments.

**Strengths:**

This paper integrates tag matching and contrastive ranking.
The analysis of long-tailed performance presented in this paper is of particular significance.

**Limitations:**

1.	The significance of Fig 1, the teaser image, is unclear; it neither reveals the structure of the entire paper nor distinguishes it from existing methods.
2.	The paper lacks consistency throughout, such as the discrepancy in some formula symbols in Fig 2 compared to those used later on.
3.	In constructing the tag libraries, DPC2022 was used, which has a broader coverage than AVA. Could you analyze potential data leakage issues?
4.	Prompt templates in equations 3 and 4 could be discussed in more detail, including a comparison of different templates.
5.	Equations 5 and 6 seem to have incorrect dimensions; the current notation suggests 'sim' is a scalar rather than a matrix.
6.	The need for prior semantic tag mining raises questions about fairness, and the paper appears not to compare with other studies that design anchor images.
7.	A clearer explanation of Balanced Sampling is recommended.
8.	The explanation for Contrastive Ranking structure could be more detailed.

**Suitability:**

3

---

### Official Review · Reviewer_4ACf · 2024-05-20

**Rating:** 3
**Confidence:** 3

**Summary:**

In the paper, authors argue that Image Aesthetics Assessment is closely related to the image semantics, so they use tag matching strategy to model the image semantics, and proposed an image aesthetics assessment method using tag matching and contrastive ranking.

**Strengths:**

This work provides a new perspective for image aesthetic assessment, that is, the impact of image semantics on IAA. They introduced tag matching and contrastive ranking into IAA.

**Limitations:**

The analysis of experimental results is not sufficient. In Tab 1, some metrics is lower than AesCLIP (TMCR Swin-T(b) vs AesCLIP ViT), but the author did not analyze the reasons.
I doubt the advancement of the method. Compared TMCR Swin-T(b) and AesCLIP ViT,
most of the metrics of the proposed method are higher, but the metrics of TMCR ViT is almost lower than AesCLIP ViT. So, with the same backbone, the effect of this method is not as good as AesCLIP.
Some details of the method are not clearly described. The diagram of model architecture is too simple. It is best to give a complete structure and indicate which parts are trainable and which parts are frozen. Is this model trained in three stages, tag matching stage, contrastive ranking stage and aesthetics prediction stage? It is better to describe clearly. The details of clustering are not clearly described, such as the number of categories.

**Suitability:**

3

---

### Official Review · Reviewer_YVZr · 2024-05-23

**Rating:** 4
**Confidence:** 3

**Summary:**

This paper proposes a semantic-aware image aesthetics assessment method, aiming to solve the challenges caused by the diversification of image semantic backgrounds in image aesthetics assessment (IAA). The method models the aesthetic differences between images from two perspectives: aesthetic attributes and aesthetic levels. Specifically, two strategies called tag matching and contrastive ranking, are proposed to extract knowledge related to image aesthetics. Tag matching identifies semantic categories and main aesthetic attributes based on a predefined tag library, and contrast ranking reveals comparative relationships between images with different aesthetic levels but similar semantic backgrounds through balanced sampling and traversal contrast learning.

**Strengths:**

1.	The proposed semantics-aware image aesthetics assessment method, using tag matching and contrastive ranking, fills the gap in existing IAA methods by distinguishing aesthetic differences within the same semantic context.
2.	By employing balanced sampling and traversal contrastive learning, the method effectively mitigates the long-tail distribution problem in aesthetic data, improving the model's prediction performance across different aesthetic levels.
3.	Experiments on benchmark databases demonstrate the competitive performance of the proposed model in terms of prediction accuracy and handling long-tail effects.

**Limitations:**

1.	CLIP's matching capability is not been fully utilized. Specifically, the way the authors input words into the text encoder to obtain text features is inconsistent with CLIP's original pre-training method using prompts (sentences). In that case, using other single-modal models might achieve similar results, as the interaction between text and vision, a key feature of CLIP, is not being leveraged.
2.	The construction of semantic and aesthetic attribute tag libraries relies on manual filtering and tools like GPT-4, increasing the complexity and cost of implementing the method. The accuracy and coverage of manually curated tags directly impact the model's performance.

**Suitability:**

3

---

### Meta-Review · Area_Chair_Zron · 2024-07-03

**Recommendation:** Accept (Poster)
**Confidence:** 4

**Metareview:**

The paper introduces a novel approach to IAA that addresses important challenges in the field. While there were some initial concerns, the authors have adequately addressed most of them in their rebuttal. The paper's strengths in tackling semantic awareness in IAA and its effective handling of long-tail distribution make it a valuable contribution to the field. The integration of tag matching and contrastive ranking strategies offers a fresh perspective on IAA that could inspire further research in this direction. Therefore, based on the reviewers' feedback and the authors' rebuttal, I recommend acceptance for this paper. I encourage the authors to carefully consider the reviewers' feedback when preparing the camera-ready version.